# IL-33 Induces an Antiviral Signature in Mast Cells but Enhances Their Permissiveness for Human Rhinovirus Infection

**DOI:** 10.3390/v14112430

**Published:** 2022-11-01

**Authors:** Charlene Akoto, Anna Willis, Chiara F. Banas, Joseph A. Bell, Dean Bryant, Cornelia Blume, Donna E. Davies, Emily J. Swindle

**Affiliations:** 1Clinical and Experimental Sciences, Faculty of Medicine, University of Southampton, University Hospital Southampton, Southampton SO16 6YD, UK; 2Cancer Sciences, Faculty of Medicine, University of Southampton, University Hospital Southampton, Southampton SO16 6YD, UK; 3Human Development and Health, Faculty of Medicine, University of Southampton, University Hospital Southampton, Southampton SO16 6YD, UK; 4NIHR Southampton Biomedical Research Centre, University Hospital Southampton, Southampton SO16 6YD, UK

**Keywords:** rhinovirus, infections, IL-33, mast cells

## Abstract

Mast cells (MCs) are classically associated with allergic asthma but their role in antiviral immunity is unclear. Human rhinoviruses (HRVs) are a major cause of asthma exacerbations and can infect and replicate within MCs. The primary site of HRV infection is the airway epithelium and MCs localise to this site with increasing asthma severity. The asthma susceptibility gene, IL-33, encodes an epithelial-derived cytokine released following HRV infection but its impact on MC antiviral responses has yet to be determined. In this study we investigated the global response of LAD2 MCs to IL-33 stimulation using RNA sequencing and identified genes involved in antiviral immunity. In spite of this, IL-33 treatment increased permissiveness of MCs to HRV16 infection which, from the RNA-Seq data, we attributed to upregulation of ICAM1. Flow cytometric analysis confirmed an IL-33-dependent increase in ICAM1 surface expression as well as LDLR, the receptors used by major and minor group HRVs for cellular entry. Neutralisation of ICAM1 reduced the IL-33-dependent enhancement in HRV16 replication and release in both LAD2 MCs and cord blood derived MCs. These findings demonstrate that although IL-33 induces an antiviral signature in MCs, it also upregulates the receptors for HRV entry to enhance infection. This highlights the potential for a gene-environment interaction involving IL33 and HRV in MCs to contribute to virus-induced asthma exacerbations.

## 1. Introduction

Mast cells (MCs) are tissue-resident immune cells that play important roles in innate and adaptive immunity. Their localisation within tissue enables them to form the first line of defence against invading pathogens, particularly parasites and bacteria [1]. However, in susceptible individuals, MCs are classically associated with the early phase reaction in allergic asthma [2,3] in which specific allergens interact with IgE to cause FcɛRI crosslinking on the surface of MCs leading to rapid release of mediators that cause bronchoconstriction. In asthma, MC numbers are increased in the bronchial epithelium, airway smooth muscle and submucosal glands and have an activated phenotype [4]. MC location and phenotype change with increasing asthma severity [5,6,7], are closely related to Th2 biomarkers [6] and MC activation is prominent in multidimensional clustering (MDS) of severe disease [8,9]. Recent studies demonstrate that MC activation signatures are enriched across multiple clinical and molecular phenotypes of severe asthma [10] providing a strong association of MCs in asthma pathogenesis.

Due to their location at mucosal surfaces, MCs are important sentinel cells involved in immunity towards parasites and bacteria [1] but their role in antiviral immunity and virus-induced exacerbations are less well defined. MCs release interferons (IFNs) following Toll-like receptor 3 (TLR3) activation, influenza A and respiratory syncytial virus (RSV) infection [11] and release chemokines and cytokines which recruit immune cells following infection with dengue virus or RSV [12,13,14]. While MCs can be infected with viruses and mount an immune response, they generally do not support replication of viruses, including RSV or influenza [14,15,16]. Human rhinoviruses (HRVs) are a major risk factor for asthma development in early life [17] and are the major cause of viral-induced exacerbations of asthma [18]. In a study of experimental HRV infection, MC numbers were increased in the airways of asthmatic subjects and correlated with improved lung function during acute infection [19] suggesting MCs may play a protective role in virus-induced exacerbations of asthma. In contrast, treatment of asthmatic patients with the anti-IgE therapy omalizumab reduced MC numbers [20] and virus-induced exacerbations [21,22] suggesting a harmful role for MCs in asthma exacerbations. We have previously shown that while MCs mount an innate immune response to HRV infection, they are permissive for viral replication and release infectious virus particles independent of cell death [23]. This is distinct amongst immune cells, which generally mount immune responses and do not release virus progeny [24,25,26]. While MCs appear to lack a robust type I IFN response to HRV infection, addition of exogenous IFN-β protects them against infection [23].

Unbiased genome wide association studies (GWAS) have identified a large number of asthma susceptibility genes (and variants) that are expressed in the airway epithelium. These genes include *IL33* and its receptor *IL1RL1* (suppression of tumorigenicity 2 (ST2)) [27,28,29,30]. The potential importance of the IL-33/ST2 axis in asthma is supported by several studies in which IL-33, ST2 and secretory ST2 (sST2) are increased in asthma compared to healthy controls and positively correlate with disease severity [31,32,33,34,35]. Furthermore, IL-33 enhances type 2 cytokine secretion by cells associated with asthma pathogenesis including MCs, basophils, T cells and type 2 innate lymphoid cells (ILC2s) [36,37]. Due to its enhancing effect on type 2 cytokine secretion, approaches to neutralise IL-33 are currently being developed for asthma therapy [38] and have shown a reduction in exacerbation rates [39]. However, the role of IL-33 in innate immunity is not fully understood.

IL-33 is an alarmin with many roles including tissue homeostasis and repair, type 2 inflammation and viral infections [40]. Depending on the type of viral infection IL-33 has either a protective or detrimental role and appears to be highly specific depending on the virus. For example, IL-33 is protective during choriomeningitis virus or herpes simplex virus infection while it exacerbates RSV and influenza virus-induced airway inflammation [41]. During experimental HRV infection, IL-33 is detected in the airways of asthmatic subjects and correlates with type 2 cytokine production, lower respiratory symptom scores and viral load [37]. In addition, in vitro HRV-infected bronchial epithelial cell (BEC) supernatants induced T cell and ILC2 type 2 cytokine release, which was suppressed following incubation with a ST2 blocking antibody suggesting that epithelial-derived IL-33 contributes to type 2 inflammation [37]. While IL-33 enhances IgE-dependent MC responses and induces type 2 cytokine secretion [42], its role in HRV-mediated MC responses is unknown.

In this study we investigated the global response of MCs to IL-33 stimulation and whether this has a functional consequence during viral infection. Following incubation of LAD2 MCs with IL-33, transcriptomic analysis revealed characteristic upregulation of type 2 cytokine genes, as well as genes involved in antiviral immunity. However, when tested functionally, IL-33 treatment caused an unexpected increase in MC infection by HRV16; this was shown to be caused by an IL-33-dependent increase in intercellular adhesion molecule 1 (ICAM1), the cellular receptor used for HRV16, a major group rhinovirus. These effects were confirmed in both LAD2 MCs and cord blood derived MCs (CBMCs). The receptor used by minor group HRVs, low-density lipoprotein receptor (LDLR) was also upregulated by IL-33 treatment. The impact of IL-33 on MCs may have important consequences in viral-induced exacerbations of asthma: IL-33 release by a virally infected epithelium may predispose MCs for increased infectivity by HRV allowing MCs to contribute in a detrimental way to rhinovirus induced exacerbations of asthma.

## 2. Materials and Methods

Detailed methods can be found in Appendix A.

### 2.1. Mast Cell Culture

The human MC line, LAD2, was obtained from Dr. A. Kirshenbaum (National Institutes of Health, Bethesda, USA) [43] and was cultured as previously described [23]. Primary cord blood-derived MCs (CBMCs) were generated as previously described [23] from a commercial source of CD34+ cord blood progenitor cells (StemCell technologies, Grenoble, France). 

### 2.2. Rhinovirus Stocks

Human rhinovirus serotype 16 (HRV16, major group, HRV-B species) stocks were generated using H1-HeLa cells obtained from ATCC as previously described [23] and titres determined by tissue culture infective dose 50% (TCID_50_)/mL. 

### 2.3. Cell Treatments and Infections

Human MCs (1 × 10^6^/mL) were incubated with IL-33 (1–10 ng/mL, R&D Systems, Abingdon, UK) for 6–24 h in a humidified 37 °C incubator with 5% CO_2_ before transcriptomic and reverse transcription qPCR analyses. For HRV16 infection, MCs were pre-treated with IL-33 and then incubated with increasing multiplicity of infection (MOI: 1–7.5) of infectious virus or UV-irradiated virus (as a control) for 1 h, washed twice then resuspended in StemPro media (0.5–1 × 10^6^/mL) for specified times before harvesting. For ICAM-1 blocking experiments, LAD2 MCs or CBMCs (1 × 10^6^/mL) were treated with 10 ng/mL IL-33 for 23 h followed by 1 h with mouse anti-human ICAM-1 (clone BBIG-I1 (11C81), 10 µg/mL, R&D Systems, Abingdon, UK) or IgG2a isotype control (10 µg/mL, R&D Systems, Abingdon, UK). Experimental layouts are shown in Appendix A.

### 2.4. mRNA Extraction, RNA Sequencing and Transcriptomic Analysis

Total RNA was isolated using either the Trizol extraction method as previously described [23] or using commercially available kits (RNeasy minikit, Qiagen (Manchester, UK) or Monarch total RNA miniprep kit (New England Biolabs, Hitchin, UK)). Next generation sequencing was performed by Novogene (Cambridge, UK) before data processing and analysis as detailed in Appendix A Briefly, raw reads were mapped to the human genome (HISAT2), converted to counts (SAMtools) and adjusted for batch effects (ComBat Seq within sva package). After filtering out low counts (EdgER) remaining counts were normalised using a weighted trimmed mean of the log expression ratios (trimmed mean of M values (TMM)). The resultant expression matrix was used to create multidimensional scaling (MDS) plots (limma, Rstudio) and Heatmaps (heatmap.2, RStudio) and fitted to a generalised linear model (quasi-likelihood F-test) for differential expression. Differentially expressed genes (DEGs) were defined as genes with a log_2_(fold change) (log_2_FC) > 1.5 and a False Discovery Rate (FDR)-adjusted *p* value < 0.05. Upregulated genes were subjected to Gene ontology (GO) and pathways analysis (KEGG). Data are available at GSE216269.

### 2.5. RT-qPCR

cDNA template (12.5 ng) was used in quantitative PCR (qPCR) with Precision Plus double dye primers for housekeeping genes (HKGs; glyceraldehyde-3-phosphate dehydrogenase (*GAPDH*), ubiquitin C (*UBC*) or genes of interest (interferon induced with helicase C domain 1 (*IFIH1*)*,* interferon regulatory factor 1 (*IRF1*)*,* tumour necrosis factor alpha (*TNFA*)*,* IFN beta 1 (*IFNB1*)*,* IFN lambda 1 (*IFNL1*)*,* 2′,5′-oligoadenylate 1 (*OAS1*)*,* C-X-C motif chemokine ligand 10 (*CXCL10*)*, IL6,* C-C motif chemokine ligand 5 (*CCL5*)) or SYBR^®^ green primers for genes of interest (*ICAM1*) used to quantify amplification of genes using a real-time PCR iCycler (BioRad, Hemel Hempstead, UK). Gene expression was normalised to the geometric means of HKGs and fold changes in gene expression calculated relative to UV-HRV16 controls according to the ^ΔΔ^Ct method and expressed as 2^−ΔΔCt^. Viral RNA copy number was determined against a standard curve of known copies of HRV16 (Primerdesign, Chandlers Ford, UK).

### 2.6. Flow Cytometry

MCs (0.1 × 10^6^/100 µL) were incubated with fluorescently labelled antibodies, FITC-conjugated mouse anti-human ICAM1 (clone RR1/1), subclass IgG_1_) or mouse IgG_1_ isotype control (eBiosciences, Cheshire, UK) for 30 min with the addition of eBioscience™ Fixable Viability Dye eFluor™ 660 (Thermo Fisher Scientific, Paisley, UK) on ice prior to resuspending in 300 µL FACS buffer. Flow cytometry was performed using a BD FACSCalibur flow cytometer (BD Biosciences, Oxford, UK) and data analysed using FlowJo software (version 7.6.5, Oregon, BD, USA). 

### 2.7. TCID_50_ Assay

The number of infectious virus particles in cell-free supernatants was determined by the TCID_50_ assay where a 10-fold serial dilution of supernatants in quadruplicate were added to OHIO HeLa cells (0.2 × 10^6^/well, 96-well plate). After 96 h, cytopathic effect (CPE) was visualised by staining monolayers with crystal violet solution (0.13% (*w*/*v*), 1.825% (*v*/*v*) formaldehyde, 5% ethanol (*v*/*v*), 90% PBS (*v*/*v*) for 30 min in the dark. Excess crystal violet was removed by gentle rinsing and the number of wells where at least 50% of the monolayer had been lysed (i.e., 50% CPE) was used to calculate TCID_50_/mL using the Spearman-Karber Method.

### 2.8. Statistical Analysis

Paired non-parametric data were analysed by Wilcoxon signed rank test for matched pair comparisons. Un-paired non-parametric data were analysed by Kruskal–Wallis one-way ANOVA with Dunn’s correction for multiple comparisons or Mann–Whitney ranked sum test and normalised data were analysed by Student’s *t*-test. All data were analysed using GraphPad Prism (GraphPad Software, Inc., San Diego, CA, USA) and results were considered significant if *p* ≤ 0.05, where * *p* ≤ 0.05, ** *p* ≤ 0.01, *** *p* ≤ 0.001, **** *p* ≤ 0.0001.

## 3. Results

### 3.1. Transcriptomic Analysis of MCs Exposed to IL-33 Reveals an Antiviral Gene Signature

To determine the global response of LAD2 MCs following IL-33 stimulation, RNA-Seq was performed. Following 6 h of treatment of MCs with IL-33, 414 differentially expressed genes (DEGs) (log_2_FC >1.5, FDR-adjusted *p* < 0.05) were identified (354 upregulated and 60 downregulated) (Figure 1A). Hierarchical cluster analysis of DEGs identified 3 clusters that were upregulated in response to IL-33 treatment and two much smaller clusters that were down-regulated (Figure 1B). Of the top 100 most significantly DEGs, the majority were upregulated (Figure 1C). Further inspection of the top 50 DEGs (by log_2_FC) identified genes known to be associated with IL-33 stimulation (e.g., cytokines such as *IL13, IL5*) (Appendix A). However, this group also included genes encoding the virally induced gene Epstein–Barr Virus Induced 3 (*EBI3*) and the viral-recognition gene (*TLR7*) (Appendix A). 

Gene ontology (GO) analysis identified that IL-33 treated MCs were enriched for genes associated with response to stimulus, regulation of response to stimulus, signaling, positive regulation of biological processes, cell adhesion and regulation of cell adhesion, among many others (Figure 2A). When the list of GO processes significantly induced by IL-33 were ranked according to adjusted *p* value, genes associated with innate immunity including defense response (adj *p* value 4.81 × 10^−21^) and innate immune response (adj *p* value 4.91 × 10^−07^) were upregulated and more specifically genes associated with viral immunity including response to virus (adj *p* value 4.27 x10^−05^), cellular response to virus (adj *p* value 0.005), negative regulation of viral genome replication (adj *p* value 0.044) and defense response to virus (adj *p* value 0.046). Kyoto Encyclopedia of Genes and Genomes (KEGG) pathway analysis identified in 35 significantly upregulated pathways (Figure 2B). The top pathways associated with IL-33 activation were, as expected, ‘cytokine-cytokine receptor interactions’, ‘NF-kappa B signalling pathway’ and ‘TNF signalling pathway’ and those associated with innate immunity (16 of 35). Amongst these pathways 7 of 16 were involved in anti-viral immunity (‘viral protein interaction with cytokine and cytokine receptor’, ‘Toll-like receptor signaling pathway’, ‘coronavirus disease—COVID-19′, ‘influenza A’, ‘measles’, ‘epstein-Barr virus infection’, and ‘human cytomegalovirus infection’) (Figure 2B). Of particular interest was genes involved in the ‘influenza A’ pathway which included those involved in viral detection (IFIH1(MDA5), OAS2 and OAS3 (2-5′OAS), TLR7) and inflammation (IL1B, CXCL8 (IL8), IL33, TNFA, CCL2 (MCP1), CCL5 (RANTES), CXCL10 (IP-10)) (Figure 2C); further inspection of the DEGs also identified other antiviral genes associated with HRV infection. In total 24 antiviral genes were up-regulated by IL-33 (Appendix A) including IFIH1, and TNFA whose expression was confirmed by qPCR (Figure 2D). RTqPCR also demonstrated that induction of antiviral genes (IFNB1, IFNL1, IFIH1, OAS1, CCL5, CXCL10) occurred in an IL-33 concentration-dependent manner and persisted for up to 24 h (Appendix A). These data demonstrate that genes involved in antiviral immunity are induced in MCs following IL-33 stimulation.

### 3.2. IL-33 Induces Interferons and IFN-Stimulated Genes in LAD2 MCs

To determine the functional consequence of the upregulation of antiviral genes in MCs by IL-33, LAD2 MCs were pre-treated with IL-33 (1–10 ng/mL) for 24 h prior to infection with HRV16 (MOI 7.5) using UV-irradiated HRV16 as a control. Analysis of interferons (IFNs) and IFN stimulated genes (ISGs) 24 h post-HRV16 infection revealed induction of *IFNB1* and *IFNL1* genes by HRV16 but not UV-HRV16 (Figure 3A,B). IL-33 concentration dependently increased *IFNB1* in the absence of infection while in the presence of both IL-33 and HRV16 there was a further significant increase in expression of *IFNB1* and *IFNLI* (Figure 3A,B). A significant increase of HRV16-dependent secretion of IFN-β protein in the presence of IL-33 (10 ng/mL) was confirmed by ELISA (Figure 3E); no IFN-β protein was detected in the UV-HRV16 control in the absence or presence of IL-33. There was also an IL-33 concentration-dependent increase in the gene expression of ISGs *IFIH1* and *OAS1* in the presence of HRV16 (Figure 3C,D) suggesting that IL-33 enhanced antiviral immunity in MCs. Proinflammatory cytokine gene expression including *CXCL10* and *IL6* were similarly enhanced by IL-33 pre-treatment in both UV-HRV16 and HRV16 treated samples (Appendix A) which was confirmed at the protein level for IL6 (Appendix A). As anti-viral responses were greatest with 10 ng/mL IL-33, the following experiments were conducted at this concentration.

### 3.3. IL-33 Enhances HRV16 Replication and Release of Infectious Viral Particles through Increased ICAM-1 Expression

In view of the enhanced anti-viral response mediated by IL-33, we next investigated the impact of IL-33 on viral infection. LAD2 MCs were incubated with IL-33 for 24 h prior to HRV16 infection (MOI 7.5) for a further 24 h. RT-qPCR for the viral genome showed IL-33 significantly increased HRV16 replication, which was paralleled by a significant rise in the release of infectious virus particles as determined by TCID_50_ assay (Figure 4A,B) suggesting that rather than being protective, IL-33 promotes the infection of MCs by HRV16. 

Since IL-33 has been shown to increase expression of ICAM1, the receptor for major group HRVs, on endothelial cells [44] and murine MCs [45], we inspected the RNASeq data for expression of *ICAM1* and identified it within the DEGs (Appendix A). Flow cytometric analysis confirmed that IL-33 significantly increased the cell surface expression of ICAM1 (Figure 5A,B) after 24 h incubation. While ICAM1 is the receptor for major group HRVs, minor group HRVs bind to the LDLR and HRVC binds to CDHR3. Therefore, we investigated the effect of IL-33 on expression of these receptors. By RTqPCR, we confirmed upregulation of *ICAM1* and found no effect of IL-33 on the low expression of *CDHR3* (control Ct 34.42 ± 0.37) (Appendix A), consistent with our previous study which reported that MCs do not express CDHR3 [23]. Flow cytometric analysis demonstrated the upregulation of LDLR cell surface expression by IL-33 (Appendix A) on LAD2 MCs. IL-33 also upregulated its own receptor, membrane bound ST2 in LAD2 MCs (Appendix A). To demonstrate that the increase in ICAM1 expression was responsible for the enhancement in HRV16 infection, MCs were pre-treated with IL-33 for 24 h with and without an anti-ICAM1 antibody or isotype control prior to HRV-16 infection for a further 24 h. ICAM1 blockade significantly abrogated the IL-33-dependent increase in both HRV16 replication and release (Figure 5C,D) and this was paralleled by a significant drop in IL-33 enhanced IFN-β release (Figure 5E). These data were confirmed in primary CBMCs which showed an IL-33-dependent increase in viral copy number (Figure 6A) and ICAM1 expression (Figure 6B,C), and that both HRV16 infection and IFN-β release were abrogated by anti-ICAM1 blockade (Figure 6D,F). However, CBMCs were less responsive to IL-33 stimulation than LAD2 MCs which may be explained by their lower cell surface expression of ST2, the membrane bound receptor for IL-33 (Appendix A). These data demonstrate that IL-33 increases the expression of ICAM-1 leading to enhanced replication and release of infectious virions in MCs.

## 4. Discussion

HRV infections are a major cause of asthma pathogenesis and exacerbation. MCs co-localise to the bronchial epithelium in asthma with increasing severity placing them at the main site for HRV replication and potentially contributing to HRV-induced exacerbations of asthma. *IL33* is an asthma susceptibility gene and IL-33 is released from the bronchial epithelium during HRV infection in asthmatic subjects. In this study we investigated the global response of MCs to IL-33 stimulation and determined whether this had a functional consequence during viral infection. We demonstrated that IL-33 induced an antiviral signature in MCs but rather than providing protection against HRV infection, IL-33 increased the infection of MC by HRV by causing upregulation in ICAM1, the receptor used by HRV16 for cellular entry. These data identify a potential gene-environment interaction involving the effect of IL-33 and HRV on MC that may have important consequences in virus-induced exacerbations of asthma. 

During experimental HRV16 infection of allergic asthmatic subjects, an association was found between the number of subepithelial MCs and a lower maximum percent fall in peak expiratory flow (PEF) 4 days post-infection leading the authors to suggest that MCs are beneficial during HRV infection [19]. In contrast, anti-IgE therapy, which is known to reduce MC numbers [21], reduced asthma exacerbation rates [39] indicating that MCs play a detrimental role in virus-induced exacerbations of asthma. A major difference between these investigations was the severity of asthma studied; thus mild allergic asthmatics were used for the experimental infection studies [19], whereas the effect of anti-IgE therapy involved severe asthmatic subjects [21]. As MCs appear limited in their ability to produce sufficient Type I IFNs to protect against HRV infection [23], the selection of asthma patients may have an important influence on the outcome, since in mild asthma the epithelial antiviral Type I IFN response is greater than in severe disease [46,47]. Thus, in mild asthma, the protection offered by epithelial-derived Type I IFNs towards MCs may outweigh the negative effect of IL-33 on ICAM1 expression and viral entry whereas in severe disease, a lack of epithelial IFNs may render the MCs more vulnerable to infection. In support of this hypothesis, anti-ST2 therapy (astegolimab) reduces asthma exacerbation rates in severe asthmatic subjects.

In this paper we demonstrated that IL-33 induced an antiviral gene signature. While the focus of this paper was HRVs, MCs are also implicated in other respiratory viral infections including RSV and influenza. IL-33 is induced in the epithelium following RSV infection in asthmatic subjects [48] and infection of MCs with RSV causes an antiviral response suggesting they are beneficial in RSV-induced exacerbations of asthma. Since MCs do not support RSV replication, the influence of IL-33 in this setting might be expected to enhance antiviral immunity as any potential modulation of receptors used for entry by RSV, including ICAM1 [49] would be unlikely to contribute to replication of the virus in MCs, although this remains to be determined experimentally. While IL-33 is increased in mice exposed to influenza A [50] and exogenous IL-33 is reported to protect against mucosal influenza A infection [51], its role in enhancing antiviral immunity and effects on MCs in this setting is yet to be determined.

Free IL-33 acts as a classical cytokine by binding to target cells expressing receptors for IL-33 consisting of ST2 and Interleukin-1 receptor accessory protein (IL-1RAcP) resulting in the formation of a stable dimer of the TIR-domains of ST2 and IL-1RAcP. The TIR-dimers are a scaffold for recruitment of a series of adaptor proteins, including the pivotal adaptor, MyD88, allowing activation of the classical MyD88/IRAK/TRAF6 module and downstream activation of NF-κB, stress-activated protein kinase p38 and c-Jun N-terminal kinases (JNK), as well as extracellularly regulated kinases (ERK1/2) and other signalling pathways [52]. The mechanism by which IL-33 induces antiviral immunity is yet to be determined. It may enhance Type I signalling via interleukin-1 receptor associated kinase 1/4 (IRAK1/4) and interferon regulatory factor-3/7 pathways as these are key signalling factors in the production of Type I and III IFNs in BECs [53]. 

This paper is the first to describe the global response of human MCs to IL-33 using a transcriptomics approach. While Nagarkar et al. used gene expression microarrays to report the upregulation of cytokines, chemokines, and growth factors in IL-33 treated human peripheral blood derived MC [54], the entire dataset has not been made freely available. Related studies have been performed using mouse MCs where the global response of MCs to IgE-mediated activation was compared using bone marrow-derived MCs (BMMCs) from control and ST2 KO mice [55] This study demonstrated that the IL-33 activated transcriptome was enriched in genes commonly altered by NF-κB or in response to TNF, and a list of genes unique to IL-33 stimulation alone included some innate antiviral genes (*IRF5, CCL5*) suggesting some similarities with our human MC dataset. 

IL-33 has variable effects on MCs depending on the length of exposure, concentration used and type of stimuli and typically induces MC adhesion, proliferation, maturation and activation and secretion of mediators [42]. Short term exposure enhances IgE-mediated degranulation, leukotriene and cytokine production [36,56,57,58] while long term exposure (4 wks) suppresses IgE-mediated degranulation via downregulation of hemopoietic cell kinase [59]. Furthermore, while high IL-33 concentrations (in the ng/mL range) induce cytokine secretion and enhance IgE-mediated activation, concentrations (in the pg/mL range) render MCs insensitive to bacterial cell wall components (LPS, PGN) [60]. In our studies, we tested IL-33 at 1–10 ng/mL and found concentration-dependent effects on antiviral responses. Given their close proximity to epithelial cells in the airway of asthmatic subjects, it is possible that these concentrations may be achieved locally as a consequence of the cytopathic effect of HRVs on epithelial cells. Furthermore, other signals (e.g., allergen) present in the tissue microenvironment may make important contributions to local IL-33 concentrations to influence MC contributions to viral-induced exacerbations of asthma.

In summary, we have shown that IL-33 induces an antiviral signature in MCs. In spite of this, when tested functionally this antiviral signature was insufficient to control HRV16 replication as IL-33 increased expression of ICAM1 which facilitated infection and increased viral replication in MCs. Given that 60–80% of asthma exacerbations in adults and 90% of wheezing episodes in children [18,61,62,63] are caused by HRVs, the impact of IL-33 to promote HRV infectivity of MCs may play an important role in the worsening of asthma symptoms. As *IL33* is an important asthma susceptibility gene, this effect of IL-33 on HRV infection of MCs highlights a potential gene-environment interaction that contributes to virus-induced asthma exacerbations.

## Figures and Tables

**Figure 1 viruses-14-02430-f001:**
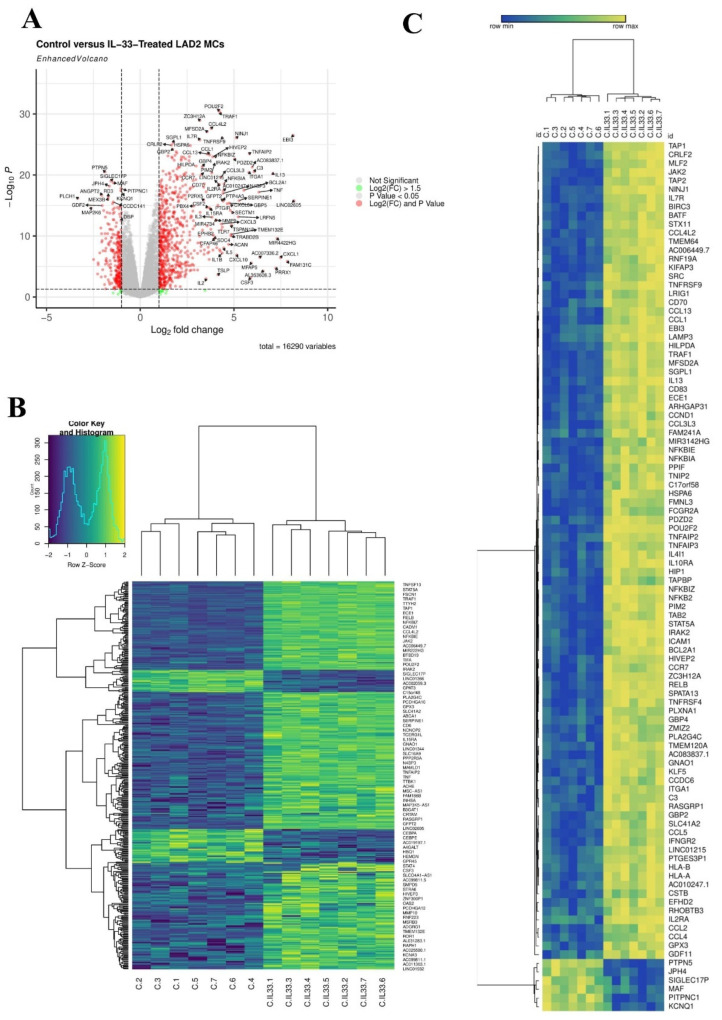
IL-33 induced 414 DEGs in MCs. (**A**)**,** volcano plot of DEGs 6 h post IL-33 (10 ng/mL) treatment, *n* = 7. (**B**), Heatmap of all 414 DEGs 6 h post IL-33 (10 ng/mL) treatment, *n* = 7. (**C**), Hierarchal clustering heatmap of top 100 most significant DEGs 6 h post IL-33 (10 ng/mL) treatment, *n* = 7.

**Figure 2 viruses-14-02430-f002:**
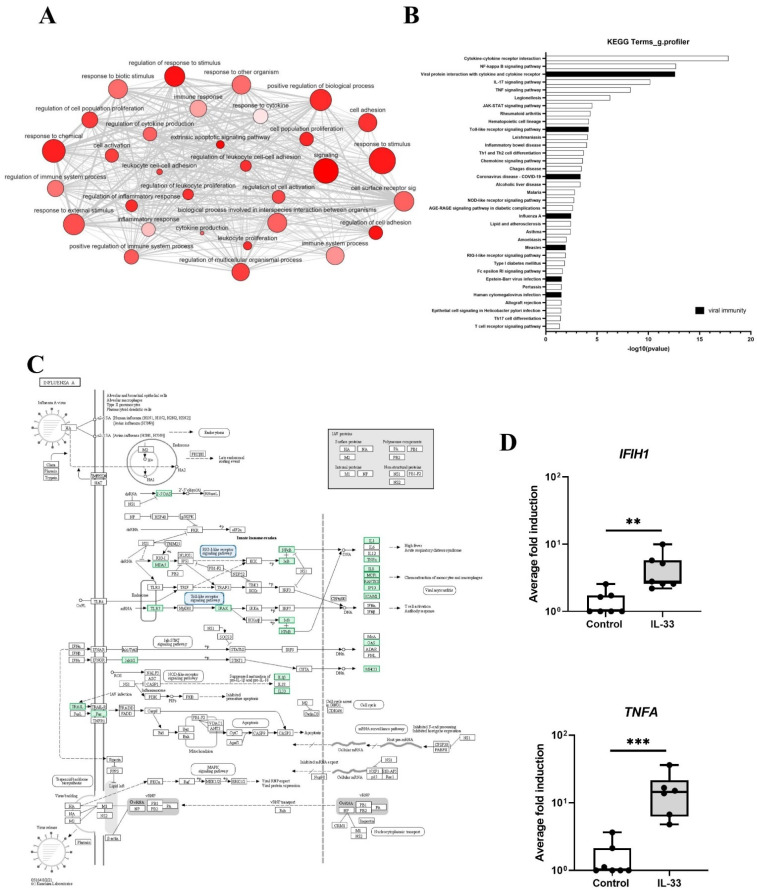
IL-33 induced DEGs associated with viral immunity in MCs. (**A**), gene enrichment map of GO terms analysed using g.profiler. (**B**), top KEGG pathways 6 h post IL-33 (10 ng/mL) treatment, *n* = 7. (**C**), genes associated with the influenza Pathway using KEGG on DEGs 6 h post IL-33 (10 ng/mL) treatment, *n* = 7. (**D**), mRNA expression of genes (*IFIH1, TNFA*) in influenza pathway 6 h post IL-33 stimulation (10 ng/mL), *n* = 7. ** *p* ≤ 0.01, *** *p* ≤ 0.001 for control versus IL-33.

**Figure 3 viruses-14-02430-f003:**
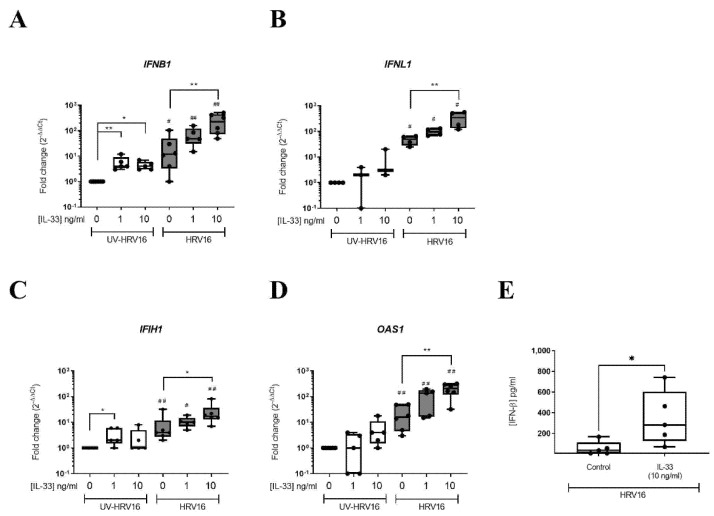
IL-33 enhanced HRV16-dependent anti-viral responses in MCs. mRNA expression of IFNs *IFNB1* (**A**), *IFNL1* (**B**) and IFN-stimulated genes *IFIH1* (**C**), *OAS1* (**D**) and IFN-β protein (**E**) in LAD2 MCs pretreated with or without IL-33 (1–10 ng/mL) for 24 h prior to HRV16 or UV-HRV16 (control) infection (MOI 7.5) for a further 24 h, *n* = 3–6. * *p* ≤ 0.05, ** *p* ≤ 0.01 for no cytokine versus IL-33 (1 or 10 ng/mL). ^#^
*p* ≤ 0.05, ^##^
*p* ≤ 0.05 for UV-HRV16 versus HRV16.

**Figure 4 viruses-14-02430-f004:**
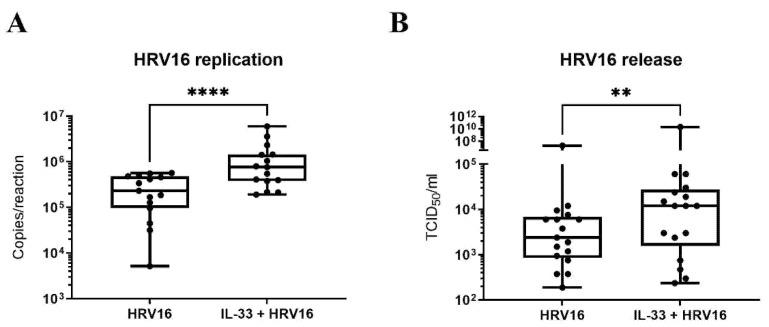
IL-33 enhances HRV16 replication and virion release in MCs. Viral RNA (**A**) and virion release (**B**) in LAD2 MCs pretreated with or without IL-33 (10 ng/mL) for 24 h prior to HRV16 infection (MOI 7.5) for a further 24 h, *n* = 15. ** *p* ≤ 0.01, **** *p* ≤ 0.0001 for HRV16 versus IL-33+HRV16.

**Figure 5 viruses-14-02430-f005:**
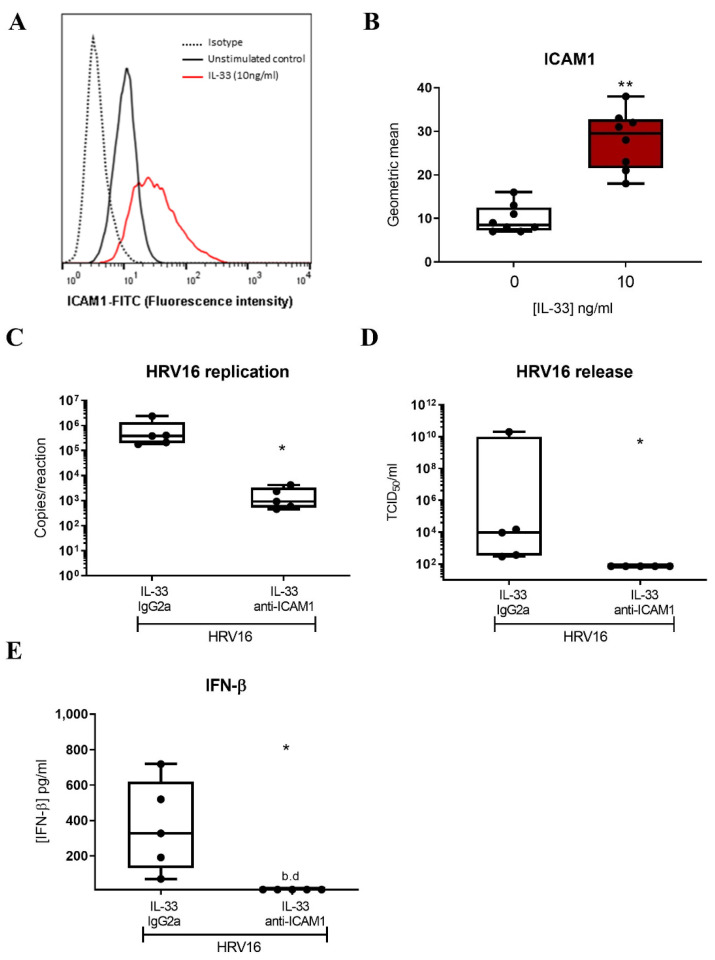
Role of ICAM1 in mediating IL-33-dependent enhancement of HRV16 replication in LAD2 MCs. A representative flow cytometric trace (**A**) and averaged geometric mean (**B**) for ICAM1 cell surface expression in MCs treated with IL-33 (10 ng/mL) for 24 h, *n* = 8. HRV16 replication (**C**) and virion release (**D**) and IFN-β protein release (**E**) following IL-33 (10 ng/mL) stimulation for 24 h prior to HRV16 infection (MOI 7.5) in the presence or absence of anti-ICAM1 antibody or IgG2a isotype for a further 24 h, *n* = 5. ** *p* ≤ 0.01 for no cytokine versus IL-33 and * *p* ≤ 0.05 for IL-33 IgG2a+HRV16 versus IL-33 anti-ICAM1+HRV16.

**Figure 6 viruses-14-02430-f006:**
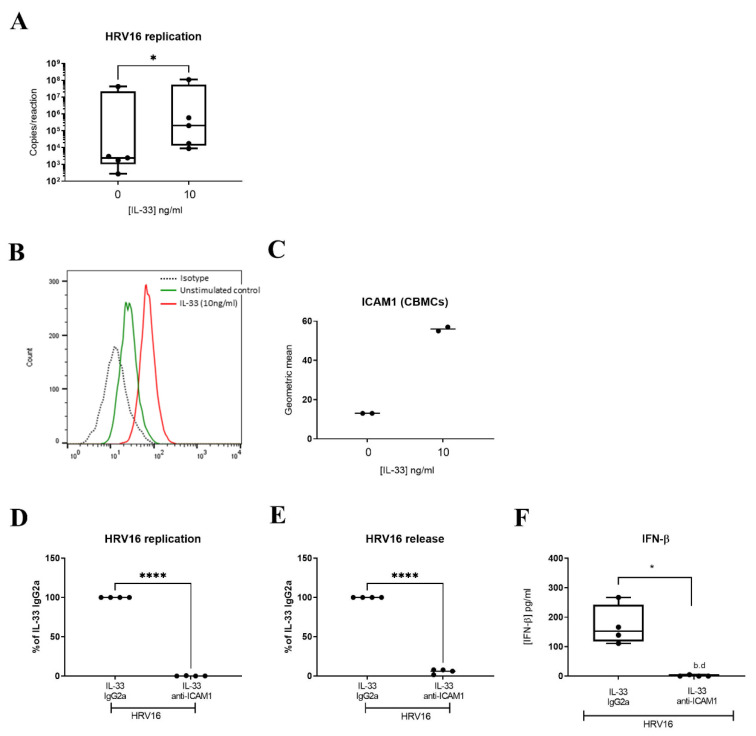
Role of ICAM1 in mediating IL-33-dependent enhancement of HRV16 replication in CBMCs. Viral RNA in CBMCs pretreated with or without IL-33 (10 ng/mL) for 24 h prior to HRV16 infection (MOI 7.5) for a further 24 h, *n* = 5 (**A**). A representative flow cytometric trace (**B**) and averaged geometric mean (**C**) for ICAM1 cell surface expression in MCs treated with IL-33 (10 ng/mL) for 24 h, *n* = 2. HRV16 replication (**D**), virion release (**E**) and IFN-β release (**F**) following IL-33 (10 ng/mL) stimulation for 24 h prior to HRV16 infection (MOI 7.5) in the presence or absence of anti-ICAM1 antibody or IgG2a isotype for a further 24 h, *n* = 4. * *p* ≤ 0.05 for no cytokine versus IL-33 and * *p* ≤ 0.05, **** *p* ≤ 0.0001 for IL-33 IgG2a+HRV16 versus IL-33 anti-ICAM1+HRV16.

## Data Availability

The data that support the findings of this study are openly available in the University of Southampton repository at https://doi.org/10.5258/SOTON/D2415 and transcriptomic data accession number GSE216269.

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
