# Peer review of "IL-33 Induces an Antiviral Signature in Mast Cells but Enhances Their Permissiveness for Human Rhinovirus Infection"

_viruses, 2022, doi:10.3390/v14112430_

Round 1
Reviewer 1 Report
Manuscript vaccines-1968628 by Akoto et al. describes the role of IL-33 in MCs responses during HRV infections. Authors describe upregulation of various genes involved in antiviral immunity including IL-33 and also describe the double edged effect of IL-33 in HRV infections: on one side, IL-33 induced IFN and IFN-stimulated genes, but on the other side IL-33 also enhanced infections of HRV in MCs which plays negative impacts on virus-induced asthma exacerbations.
This is a very well written paper, which provides highly valuable insights into IL-33 role in both anti-viral activities and virus-induced asthma exacerbations. This leads to a careful thought of IL-33 targeted regime when treating HRV infection. And this also provides insights into virus dependent outcome.
I just have minor comments:
The whole body of paper: it is recommended to provide full name of gene before use of gene symbol (e.g. ICAM1, LDLR, and CDHR3) and also to be consistent with either Italics or non-Italics when gene symbol is used. e.g. in Line 227, 228, ICAM1 in Line 227 is not italicized and the one in Line 228 is italicized.
Line 53: full name of TLR3, and RSV
Line 63: delete an extra but
Line 96: delete an extra and
Materials and Methods: it is recommended to add procedures determining TCID50 of HRV.
Line 126: incomplete sentence
Line 151, 163: full name of DEG, GO
Table A1: it is recommended to highlight the 24 genes that related to anti-viral immunity so that it aligns with Line 178
Line 171: should it be '16 of 35' instead of '16 of 34'; also 6 pathways were involved in anti-viral immunity, however in Figure 2B, 7 pathways were highlighted
Section 3.2: it is recommended to label each panel in multi-panel figure (e.g. 3A-3E), and when referring to a panel, specify the label corresponding to that panel.
Figure 3C: no INF data corresponds to UV-HRV16 condition
Figure 4, 5, 6: indicates what p value corresponds to what * note
Figure 5, 6: it is recommended to include legend to specify 'isotype', 'unstimulated control' and '10 ng/mL IL33'
Line 295: ref?
Author Response
We thank the editor and the reviewers for their valuable comments on our manuscript titled ‘IL-33 induces an antiviral signature in mast cells but enhances their permissiveness for human rhinovirus infection’. Following the opinions of the reviewers, we have further revised our manuscript. We have used the version of our manuscript found on the link provided within the email and marked any changes to the manuscript using the ‘tracked changes’ function within MS Word and have checked that all references are relevant to the contents of the manuscript. We have also tracked changes within the Appendix A and Appendix B word documents and have uploaded these to the online portal. Our point-by-point responses to the comments of the reviewer are below.
Comment 1: The whole body of paper: it is recommended to provide full name of gene before use of gene symbol (e.g. ICAM1, LDLR, and CDHR3) and also to be consistent with either Italics or non-Italics when gene symbol is used. e.g. in Line 227, 228, ICAM1 in Line 227 is not italicized and the one in Line 228 is italicized.
Response 1: We have searched the manuscript and provided the full names for genes and italicised the abbreviations.
Comment 2: Line 53: full name of TLR3, and RSV
Response 2: corrected
Comment 3: Line 63: delete an extra but
Response 3: we have reconstructed the sentence for clarity as follows (highlighted amendments in bold); ‘We have previously shown that while MCs mount an innate immune response to HRV infection, they but, despite this are permissive for viral replication and release infectious virus particles independent of cell death [23]’
Comment 4: Materials and Methods: it is recommended to add procedures determining TCID50 of HRV.
Response 4: We apologise for this oversight and have now added this method to the appropriate section. The following text has been inserted and the statistical analysis section has been renumbered 2.8. ‘2.7. TCID50 assay - The number of infectious virus particles in cell-free supernatants was determined by the TCID50 assay where a 10-fold serial dilution of supernatants in quadruplicate were added to OHIO HeLa cells (0.2x106/well, 96-well plate). After 96 hours, cytopathic effect (CPE) was visualised by staining monolayers with crystal violet solution (0.13% (w/v), 1.825% (v/v) formaldehyde, 5% ethanol (v/v), 90% PBS (v/v) for 30 minutes in the dark. Excess crystal violet was removed by gentle rinsing and the number of wells where at least 50% of the monolayer had been lysed (i.e. 50% CPE) was used to calculate TCID50/ml using the Spearman-Karber Method.’
Comment 5: Line 126: incomplete sentence
Response 5: We have amended the sentence for clarity as follows (highlighted amendments in bold); ‘For HRV16 infection, MCs were pre-treated with IL-33 and then, MCs were incubated with increasing multiplicity of infection (MOI: 1 – 7.5) of infectious virus or UV-irradiated virus (as a control) for 1 hour, washed twice then resuspended in StemPro media (0.5-1x106/ml) for specified times before harvesting’.
Comment 6: Line 151, 163: full name of DEG, GO
Response 6: corrected
Comment 7: Table A1: it is recommended to highlight the 24 genes that related to anti-viral immunity so that it aligns with Line 178
Response 7: The 24 anti-viral-related genes were found through KEGG pathway analysis and are shown as violin plots in supplementary Figure A1. The reviewer is referring to a comment about Table A1 which lists the top 50 most upregulated genes which included EBI3 and TLR7 which are linked to viral responses. We have now highlighted these 2 genes in bold in Table A1.
Comment 8: Line 171: should it be '16 of 35' instead of '16 of 34'; also 6 pathways were involved in anti-viral immunity, however in Figure 2B, 7 pathways were highlighted
Response 8: corrected and have added the name of the additional pathway which was highlighted.
Comment 9: Section 3.2: it is recommended to label each panel in multi-panel figure (e.g. 3A-3E), and when referring to a panel, specify the label corresponding to that panel.
Response 9: We have amended all multi-panel figures (Figure 3-6) and made reference to each sub-panel in the main body of the text for clarity.
Comment 10: Figure 3C: no INF data corresponds to UV-HRV16 condition
Response 10: We did not include the UV RV16 data as IFN-beta levels were below the limit of detection of the ELISA. We have now added the following phrase in the results section ‘no IFN-β protein was detected in the UV-HRV16 control in the absence or presence of IL-33 (data not shown)’.
Comment 11: Figure 4, 5, 6: indicates what p value corresponds to what * note
Response 11: We have added a sentence to the statistical analysis section in the materials and methods as follows and results were considered significant if p≤0.05, where *p≤0.05, ** p≤0.01, *** p≤0.001’. For clarity we have also provided details of the statistical analyses in the figure legends.
Comment 12: Figure 5, 6: it is recommended to include legend to specify 'isotype', 'unstimulated control' and '10 ng/mL IL33'
Response 12: we have clarified in the figure legend what each of the different histogram lines are in Figure 5A and Figure 6B.
Comment 13: Line 295: ref?
Response 13: We have added the correct reference to line 294 and updated the reference list.
Reviewer 2 Report
This study by Akoto et al. demonstrates that IL-33 enhances an antiviral response in mast cells but also increased the ability to get infected by upregulation of ICAM1. These data identify a potential dual effect of IL-33 on mast cells that may have important consequences in virus-induced exacerbations. This study is well performed and well structured. It provides novel information regarding the role of mast cells in viral infections and exacerbation as well as mast cell response to IL-33.
1. The manuscript could benefit from a figure of the experimental layout/time line to show in which orders stimulations and infections were made.
2. Since multiple timepoints and viral MOI was tested in the materials and methods, please clarify this in the results and figure legends.
3. In the material and methods, both LAD2 and primary cord blood derived mast cells were used. It is not clear from the results if both cultures were used in parallel and if there was any difference in response between the cultures. Most of the results are only showed from LAD2. Please clarify.
4. On page 5 line 96: the sentence is truncated and words seem to be missing.
5. Did the authors observe any differences in the release of chemokines, cytokines or anti-viral response depending on the time after stimulation? Does this information give indications if some mediators are pre-stored or newly produced from the mast cells?
6. The manuscript would benefit greatly from measurements of the IL-33 receptor ST2 (both membrane bound mST2 and soluble sST2) at baseline and after stimulations both with IL-33 and virus in the different mast cell cultures used.
Author Response
We thank the editor and the reviewers for their valuable comments on our manuscript titled ‘IL-33 induces an antiviral signature in mast cells but enhances their permissiveness for human rhinovirus infection’. Following the opinions of the reviewers, we have further revised our manuscript. We have used the version of our manuscript found on the link provided within the email and marked any changes to the manuscript using the ‘tracked changes’ function within MS Word and have checked that all references are relevant to the contents of the manuscript. We have also tracked changes within the Appendix A and Appendix B word documents and have uploaded these to the online portal. Our point-by-point responses to the comments of the reviewer are below.
Reviewer 2
Comment 1: The manuscript could benefit from a figure of the experimental layout/timeline to show in which orders stimulations and infections were made.
Response 1: We have added this experimental layout to Appendix B (Figure B1) detailing the timeline of stimulations and sample collections for each type of experiment.
Comment 2: Since multiple timepoints and viral MOI was tested in the materials and methods, please clarify this in the results and figure legends.
Response 2: We have added clarified the timepoints for IL-33 stimulation and HRV16 infection both within the results section and figure legends where it was omitted (Figure 5 and 6).
Comment 3: In the material and methods, both LAD2 and primary cord blood derived mast cells were used. It is not clear from the results if both cultures were used in parallel and if there was any difference in response between the cultures. Most of the results are only showed from LAD2. Please clarify.
Response 3: LAD2 MCs and primary CBMCs were used in independent experiments that were performed separately, not in parallel. Overall CBMCs were less responsive to IL-33 stimulation and this may have been due to their lower expression of ST2. We have added a sentence to the results section reflecting this observation ‘However, CBMCs were less responsive to IL-33 stimulation than LAD2 MCs which may be explained by their lower cell surface expression of ST2, the membrane bound receptor for IL-33 (Figure A5A-B)’.
Comment 4: On page 5 line 96: the sentence is truncated and words seem to be missing.
Response 4: Reviewer 1 highlighted the same issue with the sentence. We have amended the sentence for clarity as follows (highlighted amendments in bold); ‘For HRV16 infection, MCs were pre-treated with IL-33 and then, MCs were incubated with increasing multiplicity of infection (MOI: 1 – 7.5) of infectious virus or UV-irradiated virus (as a control) for 1 hour, washed twice then resuspended in StemPro media (0.5-1x106/ml) for specified times before harvesting’.
Comment 5: Did the authors observe any differences in the release of chemokines, cytokines or anti-viral response depending on the time after stimulation? Does this information give indications if some mediators are pre-stored or newly produced from the mast cells?
Response 5: This is an interesting question, however we only investigated the release of mediators after 24h of HRV-16 stimulation.
Comment 6: The manuscript would benefit greatly from measurements of the IL-33 receptor ST2 (both membrane bound mST2 and soluble sST2) at baseline and after stimulations both with IL-33 and virus in the different mast cell cultures used.
Response 6: We measured the membrane bound IL-33 receptor (mST2) but not the soluble form (sST2) following 24 hours of IL-33 stimulation in both LAD2 MCs and CBMCs. Unfortunately we did not measure the IL-33 receptor after HRV16 infection. Our result for IL-33 stimulation are presented in Appendix A, Figure A5 and show that IL-33 enhances expression of its own receptor in both LAD2 MCs and CBMCs. We have added the following sentence to the results section; ‘IL-33 also upregulated its own receptor, membrane bound ST2 in LAD2 MCs (Figure A5)’.
Reviewer 3 Report
The manuscript “IL-33 induces an antiviral signature in mast cells but enhances their permissiveness for human 2 rhinovirus infection” is an interesting study evaluated the possible role of IL-33 in anty-viral mast cell response. The study is well planned and described. The results are clearly presented, the conclusions are properly formed. I have got only minor remarks:
1. RNA-Seq and data analysis from transcriptomic measurement should be little more described in the methodology of main document
2. Figure 5a and 6b please include legend for color lines (FACS results)
3. I am not sure if results presented for n=2 (Fig 6b) should be shown in main manuscript, preferably moved to supplementary file.
Author Response
We thank the editor and the reviewers for their valuable comments on our manuscript titled ‘IL-33 induces an antiviral signature in mast cells but enhances their permissiveness for human rhinovirus infection’. Following the opinions of the reviewers, we have further revised our manuscript. We have used the version of our manuscript found on the link provided within the email and marked any changes to the manuscript using the ‘tracked changes’ function within MS Word and have checked that all references are relevant to the contents of the manuscript. We have also tracked changes within the Appendix A and Appendix B word documents and have uploaded these to the online portal. Our point-by-point responses to the comments of Reviewer 3 are below.
Comment 1: RNA-Seq and data analysis from transcriptomic measurement should be little more described in the methodology of main document
Response 1: We have described the RNA-Seq and data analysis in greater detail in the methods section and added the following text ‘Briefly, raw reads were mapped to the human genome (HISAT2), converted to counts (SAMtools) and adjusted for batch effects (ComBat Seq within sva package). After filtering out low counts (EdgER) remaining counts were normalised using a weighted trimmed mean of the log expression ratios (trimmed mean of M values (TMM)). The resultant expression matrix was used to create multidimensional scaling (MDS) plots (limma, Rstudio) and Heatmaps (heatmap.2, RStudio) and fitted to a generalised linear model (quasi-likelihood F-test) for differential expression. Differentially expressed genes (DEGs) were defined as genes with a log2(fold change) (log2FC)>1.5 and a False Discovery Rate (FDR)-adjusted p value <0.05. Upregulated genes were subjected to Gene ontology (GO) and pathways analysis (KEGG).’
Comment 2: Figure 5a and 6b please include legend for color lines (FACS results)
Response 2: Reviewer 1 highlighted the same issue. We have clarified in the figure legend what each of the different histogram lines are in Figure 5A and Figure 6B.
Comment 3: I am not sure if results presented for n=2 (Fig 6b) should be shown in main manuscript, preferably moved to supplementary file.
Response 3: For ease of comparison with the LAD2 cell data we would prefer that the cord blood derived mast cell data be kept in the main manuscript since there is such a clear effect of IL-33 on ICAM-1 cell surface expression.